# Low-voltage ultrafast nonvolatile memory via direct charge injection through a threshold resistive-switching layer

Yuan Li[1,4], Zhi Cheng Zhang[1,4], Jiaqiang Li[2,3], Xu-Dong Chen [1] ✉, Ya Kong[2], Fu-Dong Wang[1], Guo-Xin Zhang[1], Tong-Bu Lu [1] ✉ & Jin Zhang [2] ✉

The explosion in demand for massive data processing and storage requires revolutionary memory technologies featuring ultrahigh speed, ultralong retention, ultrahigh capacity and ultralow energy consumption. Although a breakthrough in ultrafast floating-gate memory has been achieved very recently, it still suffers a high operation voltage (tens of volts) due to the Fowler–Nordheim tunnelling mechanism. It is still a great challenge to realize ultrafast nonvolatile storage with low operation voltage. Here we propose a floating-gate memory with a structure of $MoS_2$/hBN/$MoS_2$/graphdiyne oxide/ $WSe_2$, in which a threshold switching layer, graphdiyne oxide, instead of a dielectric blocking layer in conventional floating-gate memories, is used to connect the floating gate and control gate. The volatile threshold switching characteristic of graphdiyne oxide allows the direct charge injection from control gate to floating gate by applying a nanosecond voltage pulse (20 ns) with low magnitude (2 V), and restricts the injected charges in floating gate for a long-term retention (10 years) after the pulse. The high operation speed and low voltage endow the device with an ultralow energy consumption of 10 fJ. These results demonstrate a new strategy to develop next-generation high-speed low-energy nonvolatile memory.

The rapid development of memory technologies including random-access memory and flash memory are indispensable for the success of information age in the past decades[1]. Nowadays, with the advent of the big data era, huge amounts of data are created annually, urgently requiring high-speed and low-energy processing and storage[2]. Current mainstream memory based on silicon technology is suffering unprecedented challenges including slow operation speed, limited storage capacity, and high energy consumption[2–6]. Revolutionary memory technologies with ultrahigh speed, ultralong retention, ultrahigh capacity, and ultralow energy consumption based on new principles,

new materials, and new structures are highly demanded[4,6–11]. In the past decade, memristors, including nonvolatile resistive switching (RS) and volatile threshold switching (TS) based on filamentary switching[12–17], have attracted wide attention due to their high operation speed and high integration density. However, the relatively large cycle-to-cycle and device-to-device variation of memristors still limit their practical application in memory. For floating-gate memory, very recent investigations have demonstrated that the atomically sharp interface of two-dimensional (2D) materials facilitates the Fowler–Nordheim (FN) tunneling of charges through the tunneling layer, demonstrating a

[1]MOE International Joint Laboratory of Materials Microstructure, Institute for New Energy Materials and Low Carbon Technologies, School of Material Science and Engineering, Tianjin University of Technology, Tianjin 300384, China. [2]Center for Nanochemistry, Beijing Science and Engineering Center for Nanocarbons, Beijing National Laboratory for Molecular Sciences, College of Chemistry and Molecular Engineering, Peking University, Beijing 100871, China. [3]Advanced Membranes and Porous Materials Center, Physical Sciences and Engineering Division, King Abdullah University of Science and Technology, Thuwal 23955-6900, Saudi Arabia. [4]These authors contributed equally: Yuan Li, Zhi Cheng Zhang. ✉e-mail: chenxd@email.tjut.edu.cn; lutongbu@tjut.edu.cn; jinzhang@pku.edu.cn

nanosecond operation speed[18,19]. Unfortunately, a high operation voltage (tens of volts) is demanded in these devices due to the FN tunneling mechanism, which will consume more energy and restrict their compatibility in complementary-metal-oxide-semiconductor (CMOS). Low-voltage ultrafast nonvolatile memory is still a great challenge.

New strategies with novel device structures are urgently needed to break the limitation of FN tunneling mechanism. For example, a semi-floating-gate memory with quasi-nonvolatile behavior was proposed[6,9,20], in which charges can be directly injected into the floating gate through a p-n junction instead of the FN tunneling, and thus it features an operation voltage of a few volts. Recently, we developed an asymmetric ultrafast nonvolatile memory based on direct-charge-injection mechanism, featuring ultrahigh speed (8 ns) and ultralow voltage (30 mV) in the writing operation[21]. Although these devices still feature some significant deficiencies, e.g., limited retention time and slow erasing speed, it provides a possible strategy to develop low-voltage ultrafast nonvolatile memory beyond the FN tunneling mechanism.

In this work, we have proposed a floating-gate memory utilizing a TS layer to connect the control gate and floating gate. The TS layer enables the direct injection of charges into the floating gate from control gate for the ultrafast writing/erasing operations (20 ns), and restricts the injected charges in the floating gate for the long retention (10 years). In particular, a quite low voltage (2 V) can be used to operate the memory device, which is over one order of magnitude lower than that of conventional floating-gate flash memories, enabling an ultralow energy consumption of 10 fJ. In addition, the high on/off ratio ($10^7$) and nanosecond-order operation speed enable the memory device to realize a 3-bit storage within a few nanoseconds. This device meets the requirements of ultrahigh speed, ultralong retention, ultrahigh capacity, and ultralow energy consumption for the next-generation memory technology.

## Results

### Architecture of the low-voltage ultrafast nonvolatile memory

The schematic structure of the ultrafast nonvolatile memory based on 2D van der Waals (vdWs) heterostructures is illustrated in Fig. 1a. The top $MoS_2$ layer serves as the channel, and a thick hexagonal boron nitride (hBN) film is used as the blocking layer. A $WSe_2$/graphdiyne oxide (GDYO)/$MoS_2$ heterostructure is located at the bottom of the device, where $WSe_2$ serves as the control gate and the $MoS_2$ layer situated under the channel and blocking layer functions as the floating gate. A TS layer, GDYO, is used to connect the control gate ($WSe_2$) and floating gate ($MoS_2$). Thus the memory device can be regarded as an integration of a transistor and a TS device (Fig. 1b). $MoS_2$, $WSe_2$, and hBN were mechanically exfoliated from bulk materials (SixCarbon Technology Shenzhen). Large-scale graphdiyne (GDY) film was synthesized via an electric-double-layer-confined strategy[22], followed by UV-ozone treatment to form GDYO (Supplementary Figs. 1–9). Details for the synthesis of GDYO film and device fabrication are described in the Experimental details in Supplementary Information. The aberration-corrected scanning transmission electron microscope (STEM) cross-sectional images and corresponding electron energy loss spectroscope (EELS) for different regions of the device are presented in Fig. 1c and Supplementary Fig. 11, demonstrating the atomically sharp interfaces of the as-fabricated heterostructures. Figure 1d presents the false-color scanning electron microscope (SEM) image of the as-fabricated memory device on $SiO_2$/Si substrate. The thickness of the top $MoS_2$, hBN, bottom $MoS_2$, GDYO, and $WSe_2$ were measured by atomic force microscope (AFM), which were 1.4 nm, 9.2 nm, 4.2 nm, 10.8 nm, and 3.6 nm, respectively (Supplementary Fig. 12). Here a bilayer $MoS_2$ was used as the channel due to its better electrostatic control, smaller bandgap and higher mobility than a monolayer, which can balance device performance and power consumption[23]. Considering that a thicker $MoS_2$ can store more charges owing to its larger density of states, while a thinner floating gate can significantly suppress the parasitic capacitive coupling between the floating gates of adjacent cells[24], we chose a 4 nm thick $MoS_2$ as the floating gate to balance these two factors.

### Threshold switching device with nanosecond switching speed

In conventional floating-gate memories, a high voltage is always required to drive charges tunneling through the blocking layer via FN mechanism[10,18,19,25–28]. To achieve ultrafast writing and erasing operations under a low voltage, our device utilizes a new strategy for charge injection, where charges are directly injected into the floating gate through a TS layer from the control gate. The conductive state of the TS layer is controlled by the electric field between two electrodes.

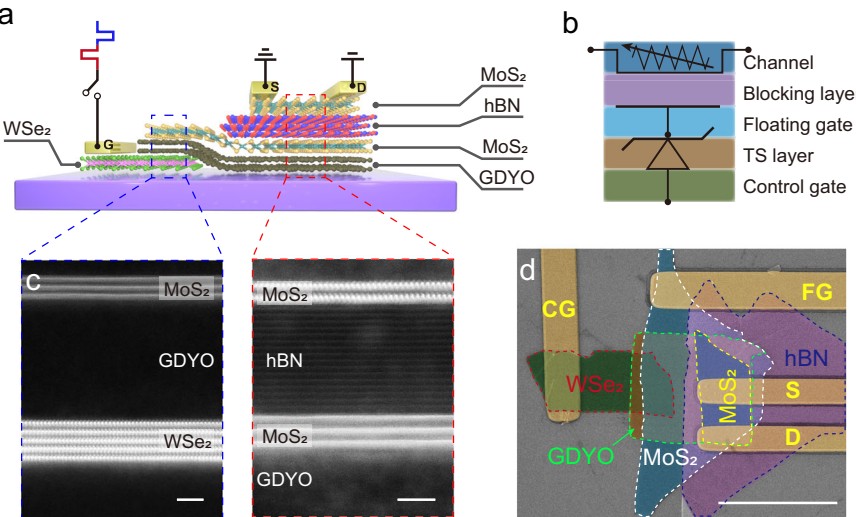

**Fig. 1 | Architecture of the low-voltage ultrafast nonvolatile memory based on 2D vdWs heterostructures. a** Schematic structure of the memory device. **b** Equivalent circuit of the memory device. The top $MoS_2$, hBN, bottom $MoS_2$, GDYO, $WSe_2$ serve as the channel, blocking layer, floating gate (FG), threshold switching (TS) layer and control gate (CG), respectively. **c** High-resolution high-angle annular dark-field (HAADF) STEM images of the $MoS_2$/GDYO/$WSe_2$ (left) and $MoS_2$/hBN/$MoS_2$/GDYO (right) heterostructures acquired from different regions of the same heterostructure as illustrated in **a**. Scale bars, 2 nm. **d** False-color SEM image of the as-fabricated memory device on $SiO_2$/Si substrate. Scale bar, 20 μm.

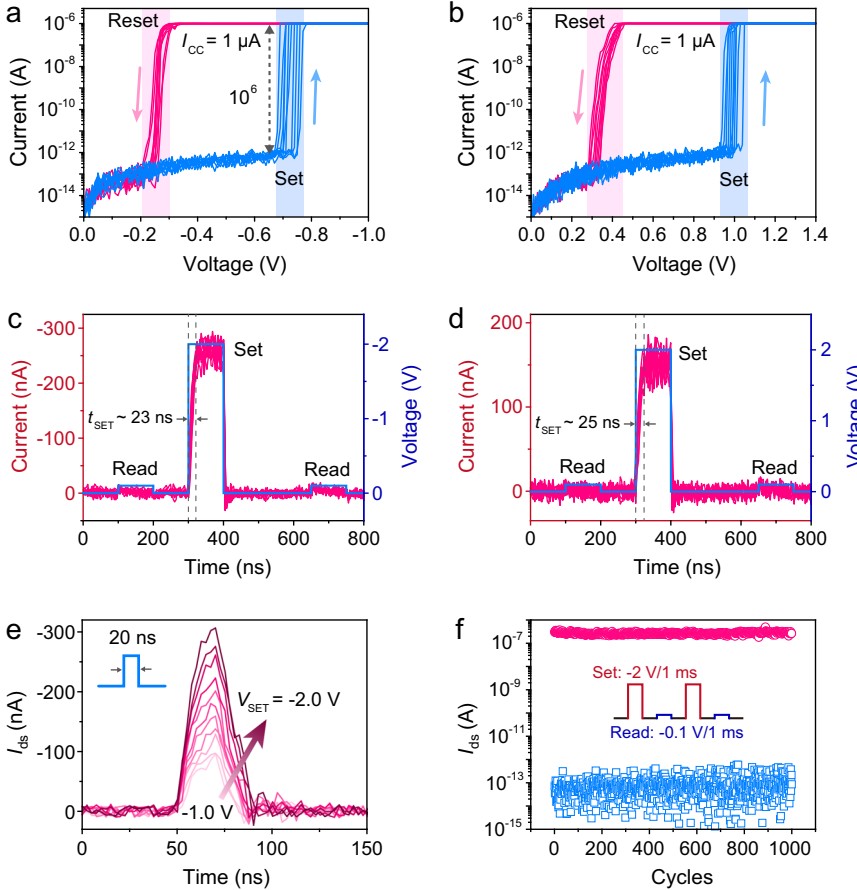

**Fig. 2 | Threshold switching device with a structure of WSe₂/GDYO/MoS₂.** $I-V$ curves of the TS device in negative (**a**) and positive (**b**) voltage regions, respectively, demonstrating a typical TS behavior. Compliance current ($I_{CC}$) was set as 1 μA. The sweeping voltage was applied to the WSe₂ terminal and the MoS₂ terminal was grounded. Transient responses of the TS device to −2 V/100 ns (**c**) and +2 V/100 ns (**d**) voltage pulses. The switching time measured in **c** and **d** are 23 ns and 25 ns, respectively. ±0.1 V/100 ns voltage pulses were applied for read operations before and after the set operation. Each experiment in **a**–**d** was repeated ten times for more reliable results. **e** Response of the TS device to 20 ns voltage pulses with amplitude ranging from −1 V to −2 V. **f** Endurance test of the TS device showing over 1000 cycles of operation. The inset illustrates the applied periodic waveforms, which consist of a −2 V/1 ms set pulse followed by a −0.1 V/1 ms read pulse, with an interval of 1 ms. The pink and blue signs represent the extracted set and read currents in each cycle, respectively.

While the applied voltage exceeds the SET voltage ($V_{SET}$), conductive filaments are formed in the TS layer, and thus charge injection from control gate to floating gate is allowed. At the termination of the voltage pulse, the conductive filaments (CFs) break and the TS device returns to its initial off-state spontaneously, preventing the escape of the injected charges[29]. Here the TS device is based on WSe₂/GDYO/MoS₂ stack, where the GDYO film is used as the active layer.

The switching behaviors of the TS device were first investigated using the circuit as illustrated in Supplementary Fig. 13a. Figure 2a presents the typical $I-V$ curves of the device in negative-voltage region. The current dramatically increases to $10^{-6}$ A at $V_{SET} = -0.72$ V (Supplementary Fig. 14) when the applied voltage sweeps from 0 V to −1 V, and the current spontaneously relaxes back to $10^{-13}$ A at around −0.25 V during the subsequent backward sweeping, demonstrating a volatile threshold switching mode[16] with a high on/off ratio of $10^6$. Due to the asymmetric band diagrams of the WSe₂/GDYO/MoS₂ (Supplementary Fig. 13b), a slightly larger SET voltage (-1 V) is required to switch the TS device from off-state to on-state (Fig. 2b). Supplementary Fig. 14 illustrates the mechanism for the switching of the device conductance. While applying a bias voltage on the device, oxygen-containing groups in GDYO migrate and form GDY CFs, resulting in the transition of the device from high-resistance state (HRS) to low-resistance state (LRS), which is similar to the SET process of memristors based on graphene oxide (GO)[30–32]. However, the difference in the structure of GDY and graphene endows GDYO- and GO-based devices with distinct resistive

switching characteristics. For the GDYO-based device, the ordered porous structure of GDY enables the diffusion of oxygen-containing groups back to oxygen-deficient regions and rupture the GDY CFs. Thus the device returns to its initial HRS spontaneously after removing the bias voltage, demonstrating a voltage TS behavior. The formation and rupture of GDY CFs can be demonstrated by the Raman mapping and conductive AFM as shown in Supplementary Figs. 16 and 17. In contrast, for the GO-based memristor, the dense honeycomb structure of graphene leads to a large diffusion barrier for oxygen-containing groups, and the migrated oxygen-containing groups are hard to diffuse back to their initial regions spontaneously, leading to nonvolatile RS or even write-once-read-many characteristics[30–32]. As discussed below, the connection between the control gate and floating gate should be ruptured to restrict the escape of injected electrons, and thus the GDYO layer with volatile TS behavior is more suitable for our memory device. Supplementary Figs. 18–21 systematically studied the influence of the GDYO area, thickness, oxygen content, and temperature on the switching behaviors of the device.

It has been demonstrated that the switching time for memristive device lies in the order of nanoseconds[15,17,33], and the switching transient for our GDYO-based TS device was measured as ~25 ns (Fig. 2c, d), indicating that a nanosecond-order voltage pulse can switch the device from off-state to on-state. The energy consumption for set transition can be calculated as $E_{SET} = I_{LRS} \times V_{SET} \times t_{SET} = 13$ fJ[15]. To investigate the voltage-dependent response of the TS device, 20 ns voltage pulses

with amplitude ranging from −1 V to −2 V were applied to the device. As shown in Fig. 2e, with the increase of the amplitude of the pulse, the on-state current of the TS device increases significantly. It is worth noting that the TS device cannot be switched on when the amplitude of the voltage is lower than 0.7 V (Supplementary Fig. 22). The robust endurance of the GDYO-based TS device has been demonstrated by applying periodic voltage pulses, which exhibits no degradation even after 1000 cycles (Fig. 2f and Supplementary Fig. 23).

**Low-voltage ultrafast nonvolatile memory device**
The nanosecond-order switching time and small SET voltage of the GDYO-based TS device enable our memory device to achieve ultrafast writing/erasing operations with low operation voltage. −2 V and +2 V voltage pulses with full-width and half-maximum of 20 ns were applied to the control gate of the device for the writing and erasing operations, respectively. As discussed in Supplementary Figs. 24 and 25, most of the applied gate voltage is loaded on the TS device, ensuring the switching of the TS device from off-state to on-state under such pulsed voltages. As shown in Fig. 3a, when a negative $V_{CG}$ pulse is applied to

the device, the MoS$_2$ channel is programmed to a low-conductance state ($10^{-13}$ A), that is, state-0. And by applying a positive $V_{CG}$ pulse, the memory can be erased to a high-conductance state ($10^{-6}$ A), that is, state-1, demonstrating a large state-1/state-0 ratio of $10^7$. The energy consumption for the ultrafast writing/erasing operations can be calculated as $E = V_{CG} \times I_{CG} \times \Delta t = 2\,V \times 2.5 \times 10^{-7}\,A \times 2 \times 10^{-8}\,s = 10\,fJ$. Figure 3b depicts the output curves of the memory device at state-1 and state-0, respectively, which were measured at $V_{CG} = 0$ V. It is worth noting that the readout operation ($V_{ds} = 0.1$ V) is nondestructive to the storage performance due to the thick hBN blocking layer used in our device.

The response speed of the memory device to the pulsed gate voltage is also crucial for the realization of ultrafast writing and erasing operations. Figure 3c, d depicts the transient responses of the device under ultrashort $V_{CG}$ pulses (20 ns) for writing and erasing operations, respectively. Here the response time $t_s$ is defined as the time that the channel current reach the 1/$e$ value of its desired state after the peak of the ultrashort pulse[18], which are 29 ns and 35 ns for the writing and erasing operations, respectively. The nanosecond-order response time

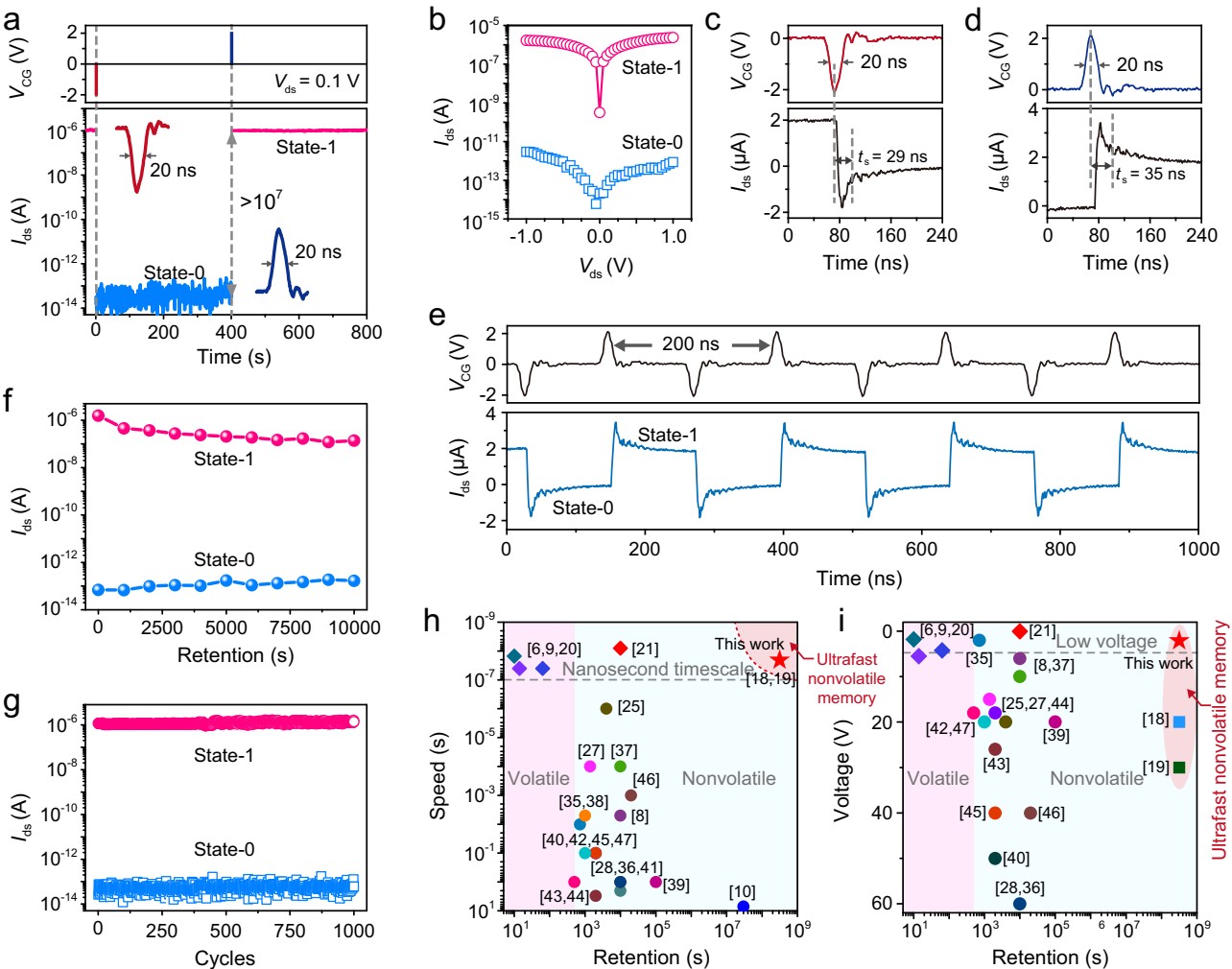

**Fig. 3 | Performance of the low-voltage ultrafast nonvolatile memory.**
**a** Ultrafast writing and erasing operations of the device. The readout operation at $V_{ds} = 0.1$ V and $V_{CG} = 0$ V demonstrated a large state-1/state-0 ratio of over $10^7$. The insets present the actual writing/erasing $V_{CG}$ pulses with FWHM of 20 ns and amplitude of −2 V and 2 V, respectively. **b** Output curves of the memory at state-1 (pink) and state-0 (blue), respectively. Transient current responses of the device by applying −2 V/20 ns (**c**) and +2 V/20 ns (**d**) $V_{CG}$ pulses for ultrafast writing and erasing operations. **e** Demonstration of ultrahigh-frequency operation by

alternatively applying ±2 V/20 ns $V_{CG}$ pulses with an interval of 100 ns. **f** Retention behaviors of the device at state-0 and state-1 after writing and erasing operations. **g** Endurance test of the memory device showing over 1000 cycles of writing/erasing operations by alternatively applying ±2 V/20 ns $V_{CG}$ pulses. **h, i** Comparison of operation speed, retention time, and operation voltage of the reported memory devices based on 2D vdWs heterostructures. The numbers in square brackets represent the ref. number listed in References section.

of the memory device indicates that the stored data can be read out instantaneously after writing/reading operations, enabling the memory to achieve a truly ultrahigh-speed operation for data storage and readout. As shown in Fig. 3e, a series of negative and positive ultrashort $V_{CG}$ pulses were alternatively applied to the device, demonstrating an ultrahigh operation frequency up to 5 MHz.

Subsequently, the data-retention characteristics and cyclic endurance of the memory device were investigated. To evaluate the data-retention property of the device, we have measured the channel currents with a fixed interval of 1000 s after both writing and erasing operations (Supplementary Fig. 26). The corresponding channel currents were extracted and summarized in Fig. 3f, demonstrating a long-term data-retention capability for over $10^4$ s. Moreover, while the retention curves are extrapolated to 10-year (dashed lines in Supplementary Fig. 27), the state-1/state-0 ratio still exceeds $10^2$, making long-term data retention become feasible at room temperature. The cyclic endurance of the device was also investigated by applying alternating negative and positive $V_{CG}$ pulses. As shown in Fig. 3g, both state-1 and state-0 remain almost unchanged with negligible degradation even after 1000 writing/erasing cycles, indicating the excellent endurance of the memory device.

Supplementary Table 1 summarizes the parameters of the reported nonvolatile memory devices based on 2D vdWs heterostructures[6,8-10,18,21,25,27,28,34-46], and Fig. 3h, i compares the operation speed, retention time, and operation voltage of these devices. In comparison, our memory device exhibits great advantages in these aspects. In particular, our ultrafast nonvolatile memory features an operation voltage as low as 2 V, which is over one order of magnitude lower than that of the recently reported ultrafast nonvolatile memories based on FN tunneling mechanism (Fig. 3i)[18,19], significantly decreasing the energy consumption and improving the CMOS compatibility.

The operation mechanism of this ultrafast nonvolatile memory is illustrated in Fig. 4. For the writing operation, a negative $V_{CG}$ pulse with

nanosecond duration is applied to the control gate. Since $V_{CG}$ is larger than the SET voltage of the GDYO-based TS device, and the required switching time of the TS device lies in the range of a few nanoseconds, the applied nanosecond $V_{CG}$ pulse can switch the TS device from off-state to on-state with the formation of GDY filaments, allowing the direct injection of electrons from the control gate to the floating gate (Fig. 4a). After the ultrafast $V_{CG}$ pulse, the volatile TS device recovers to its initial off-state spontaneously with the breaking of GDY filaments, forbidding the escape of the injected electrons in the floating gate back to the control gate (Fig. 4b). Moreover, since the electron affinity of WSe$_2$ (3.6 eV) is smaller than that of MoS$_2$ (4.2 eV)[47,48], the barrier between WSe$_2$ and MoS$_2$ would also prevent the tunneling of electrons from MoS$_2$ (floating gate) to WSe$_2$ (control gate) through GDYO layer. As a result, the injected electrons are restricted in the floating gate even after removing the external electric field, which would induce a positive shift in the threshold voltage of the top MoS$_2$ channel. Thus the device is switched to HRS with a long-term data-retention capability. For the erasing operation, while applying a positive $V_{CG}$ pulse to the device, the TS device switches on, and the stored electrons in the floating gate are extracted back to the control gate (Fig. 4c). After the $V_{CG}$ pulse, the TS device switches off again, and the memory manifests LRS at $V_{CG} = 0$ V (Fig. 4d).

To demonstrate the importance of the GDYO TS layer for the exceptional performance of our memory device, some control experiments were performed. As shown in Supplementary Fig. 28, a device with a structure of MoS$_2$/hBN/MoS$_2$/WSe$_2$ was fabricated, where the bottom MoS$_2$ and WSe$_2$ contact directly. In this device, the charges are also directly injected into the floating gate (bottom MoS$_2$) from control gate (WSe$_2$), demonstrating an ultrahigh operation speed by applying a 40 ns $V_{CG}$ pulse (Supplementary Fig. 28b). However, due to the lack of barrier layer between MoS$_2$ and WSe$_2$, the escape of the injected charges from MoS$_2$ back to WSe$_2$ through the MoS$_2$/WSe$_2$ interface is unavoidable[20], leading to a short-term retention

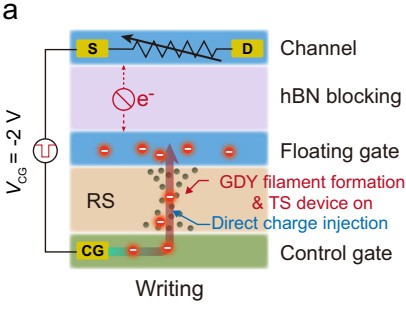

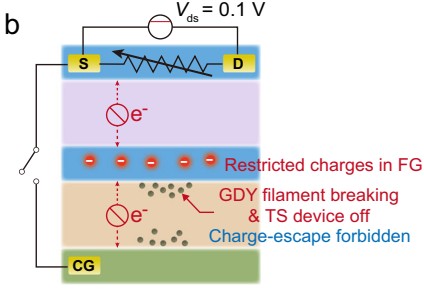

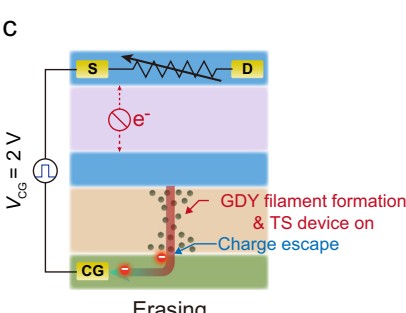

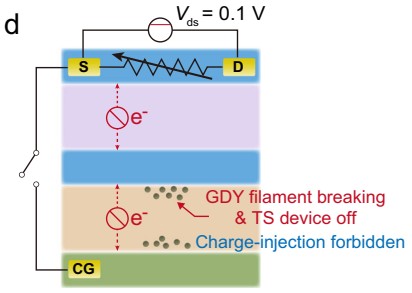

**Fig. 4 | Operation mechanisms of the low-voltage ultrafast nonvolatile memory. a** Writing operation: driven by a −2 V $V_{CG}$ pulse, GDY filaments form, and the TS device switches on, allowing the direct injection of electrons from control gate to floating gate. **b** State-0 readout: after removing the $V_{CG}$ pulse, GDY filaments breaks and the TS device switches off, forbidding the escape of injected electrons and thus restricting these electrons in floating gate. **c** Erasing operation: driven by a + 2 V $V_{CG}$ pulse, GDY filaments form and the TS device switches on, extracting the injected electrons from floating gate directly through the TS layer. **d** State-1 readout: after removing the $V_{CG}$ pulse, GDY filaments break and the TS device switches off, maintaining low-conductance state by forbidding the injection of any charges. Charges tunneling through the thick hBN blocking layer under such a low gate voltage can be ignored.

characteristic (-10 s) as previously reported[9,20]. Moreover, the thickness of the GDYO layer has a significant influence on device performance. We fabricated two memory devices with GDYO thicknesses of 3 nm and 20 nm, respectively. For the device with a 3 nm GDYO layer, although ultrafast writing/erasing operations have been demonstrated, its retention characteristics suffer serious degradation (Supplementary Fig. 29). On the contrary, the device with a 20 nm GDYO layer exhibits a long-term retention behavior, while $V_{CG}$ pulses with larger amplitude and longer duration are required for the writing/erasing operations (Supplementary Fig. 30). This can be explained by the switching behaviors of the GDYO TS layers with different thickness as shown in Supplementary Fig. 31.

Although bilayer and multilayer $MoS_2$ films are preferred to act as the channel and floating gate, respectively, it is still essential to estimate the influence of $MoS_2$ thickness on device performance, especially for monolayer grown via chemical vapor deposition (CVD). We have fabricated the memory devices using CVD-grown monolayer $MoS_2$ as the channel and floating gate, respectively, which still possess nanosecond writing/erasing speed and excellent retention characteristics (Supplementary Figs. 32 and 33). The influence of $WSe_2$ and hBN thickness on device performance is discussed in Supplementary Figs. 34 and 35, respectively.

To further demonstrate that the switching of the memory device is induced by the accumulated charges in the floating gate ($MoS_2$), rather than the trapped charges in the GDYO layer, a memory device without overlapping between the GDYO layer and the top $MoS_2$ channel was fabricated (Supplementary Fig. 36). It successively reproduced the ultrafast writing/erasing operations and long retention characteristics. Furthermore, the control gate failed to modulate the channel current while grounding the floating gate as illustrated in Supplementary Fig. 37. These results unambiguously confirm that the injected charges are accumulated in the floating gate rather than other layers.

Noteworthily, here we chose $MoS_2$ as the channel layer due to its excellent performance and stability. In fact, the channel-layer material can be extended to other 2D semiconductor materials. For instance, a p-type $WSe_2$ was used as the channel layer to construct the memory device, which successfully reproduced the ultrafast device performance with low operation voltage (Supplementary Fig. 38), highlighting the reproducibility and generality of the direct-charge-injection mechanism for low-voltage ultrafast nonvolatile memory technology.

For practical applications, large-scale device arrays with small feature sizes are required. As a demonstration, a $5 \times 5$ device array was fabricated based on CVD-grown large-area $MoS_2$ and $WSe_2$ (Supplementary Fig. 39). Noteworthily, here a 20 nm $HfO_2$ layer was used as the blocking layer to replace thick hBN. As shown in Supplementary Figs. 40 and 41, nanosecond operation speed and long-term retention have been demonstrated by all these 25 devices, which are comparable with the device fabricated by mechanical exfoliated materials. All of these 25 devices have an on/off ratio larger than $10^5$, with the largest one as $8 \times 10^6$ (Supplementary Fig. 42). Limited by the lithography equipment used in the laboratory, the channel length of the device is -2 μm. Nevertheless, we believe that nanoscale devices with high integration can be fabricated by using advanced industrial UV lithography. These results undoubtedly demonstrate the potential of our memory device for large-scale applications.

### Robustness measurement and multibit storage of the memory
The robustness of a memory device is crucial for practical applications. Here we have investigated the performance of our memory device at different temperatures and preservation time. The temperature-dependent robustness of the memory device was first investigated via measuring the device performance at temperatures in the range of 220–350 K, which covers the general ambient temperatures for

practical applications of memory devices. Supplementary Fig. 43 depicts the ultrafast writing/erasing operations of the device driven by nanosecond $V_{CG}$ pulses at different temperatures, and the corresponding channel currents of state-1 and state-0 are extracted and summarized in Fig. 5a. Supplementary Fig. 44 compares the retention characteristics of the device at different temperatures. When the temperature is larger than 330 K, the retention behavior of the device is slightly declined, since the increased temperature would enhance the leakage of charges in the floating gate. Nevertheless, the state-1/state-0 ratio still exceeds $10^3$ after $10^4$ s for the device at 350 K (Fig. 5b). The robust stability of the memory device in cold and hot environments makes it possible for practical applications in various scenarios. In addition, we have also measured the device performance with different preservation times after the fabrication of the device. As shown in Fig. 5c, d, ultrahigh-speed writing/erasing operations with comparable retention characteristics can still be demonstrated by our memory device even after 6 months (Supplementary Figs. 45 and 46), indicating a long service life of our device for practical applications.

Multibit storage can significantly improve the data-storage capacity of memory devices[7,22]. The ultrahigh operation speed and high on/off ratio of our memory device enable the device to achieve multibit storage in nanosecond timescale. By controlling the amplitude of the applied $V_{CG}$ pulse for writing operation, the amount of the injected electrons to the floating gate can be modulated (Fig. 2e), and thus the channel can achieve different conductance levels. As shown in Fig. 5e, 8 distinct storage levels with excellent retention performance have been demonstrated by applying 20 ns $V_{CG}$ pulses with amplitude ranging from −1 V to −2 V. Figure 5f exemplifies the realization of a 3-bit storage, in which the memory is first erased to 111 state by a positive nanosecond $V_{CG}$ pulse, and the other eight states of 110 to 000 can be obtained by applying a negative nanosecond $V_{CG}$ pulse with pre-designed voltage. It is worth noting that an erasing operation is required to reset the device before programming the memory to different storage levels. Benefitting from the excellent endurance of the memory device, the multibit storage exhibits good reproducibility (Supplementary Fig. 47). In addition, each storage level can be accessed independently and repeatedly by applying corresponding $V_{CG}$ pulses to the device (Supplementary Fig. 48). In comparison with the previously reported multilevel memories with a slow operation speed[7,10,22,28,38,49,50], our device exhibits unique advantages for practical applications since the multibit storage can be achieved within a few nanoseconds.

## Discussion
We have developed an ultrafast nonvolatile memory with low operation voltage. Different from conventional floating-gate flash memory, a TS layer instead of an insulator is used to connect the control gate and floating gate. The volatile TS device can switch from off-state to on-state under a small voltage stress within a few nanoseconds, and returns to initial off-state spontaneously after removing the pulsed voltage stress. Thus charges can be directly injected into the floating gate from the control gate driven by a nanosecond $V_{CG}$ pulse with a small amplitude, and these injected charges are restricted in the floating gate after the gate pulse, resulting in ultrahigh operation speed (20 ns), low operation voltage (2 V) and long retention time (10 years). Especially, the low operation voltage, which is over one order of magnitude lower than that of the conventional floating-gate flash memories, significantly decreases the energy consumption (-10 fJ) and improves the CMOS compatibility of the device. Moreover, our memory device can realize a 3-bit storage in a few nanoseconds, significantly improving the data-storage capability. This work breaks the limitation of conventional FN tunneling mechanism on device performances, providing a new strategy to develop next-generation memory technologies with ultrahigh speed, ultralong retention, ultrahigh capacity, and ultralow energy consumption.

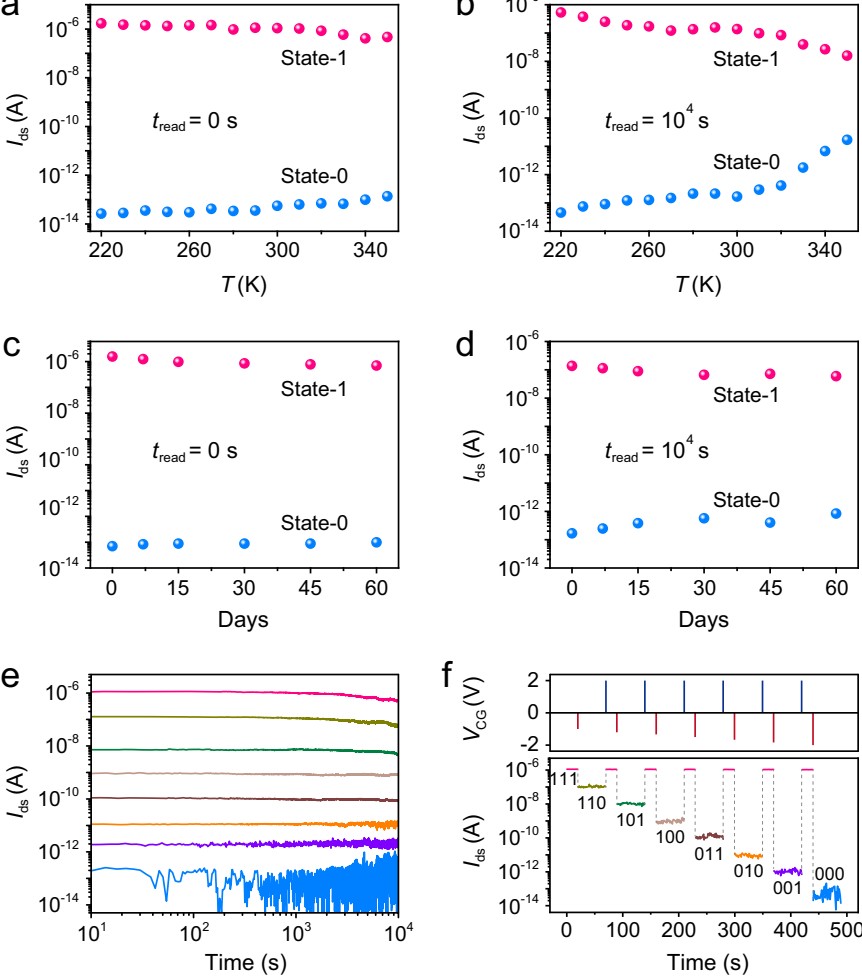

**Fig. 5 | Robustness of the low-voltage ultrafast nonvolatile memory and paradigm of multibit storage.** Variation of state-0 and state-1 currents as a function of temperature. These currents were read out immediately ($t_{read} = 0$ s) after writing/erasing operations (**a**) and after $10^4$ s retention ($t_{read} = 10^4$ s) (**b**). Variation of state-0 and state-1 currents as a function preservation time after device fabrication. These currents were read out immediately ($t_{read} = 0$ s) after writing/erasing operations (**c**) and after $10^4$ s retention ($t_{read} = 10^4$ s) (**d**). **e** $I$–$t$ curves of the memory device at eight distinct conductive levels for $10^4$ s. **f** Realization of a 3-bit (8 storage levels) memory by applying 20 ns $V_{CG}$ pulses with amplitude ranging from −1 V to −2 V. All the channel currents in **a**–**f** were read with a fixed $V_{ds}$ of 0.1 V at $V_{CG} = 0$ V.

## Methods

### STEM sample preparation and characterization
$MoS_2/hBN/MoS_2/GDYO/WSe_2$ heterostructure was fabricated on $SiO_2$/Si substrate. Then STEM samples at the $MoS_2/hBN/MoS_2/GDYO$ and $MoS_2/GDYO/WSe_2$ regions were prepared via a FEI Helios Nanolab 460HP focused-ion-beam microscope. HAADF, bright-field, and EELS imaging were performed in an aberration-corrected FEI-Titan Cubed Themis G2 300 STEM at 220 kV acceleration voltage. Raman spectroscope (Horiba Evolution) equipped with a 532 nm pump laser was used for the Raman spectra measurement.

### Device measurements
The electrical properties of the memory were measured in a probe station connected to a semiconductor characterization system (Keithley 4200A-SCS) and an arbitrary waveform generator (RIGOL DG4202) in atmospheric environment. The nanosecond $V_{CG}$ pulses were generated by the arbitrary waveform generator, while the channel current signals were monitored by the Keithley 4200 A equipped with preamplifiers. The grounds of the arbitrary waveform generator and Keithley 4200A were connected together. A current amplifier FEMTO DHPCA-100 was used to amplify the channel current, and the amplified current signals were recorded by an oscilloscope (RIGOL

DS2302A). A temperature controller equipped with the probe station was used for the temperature-dependent tests.

## Data availability
The data generated in this study have been deposited in Figshare at https://figshare.com/articles/figure/Data_for_NCOMMS-21-42325B/20355243.

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

## Acknowledgements

This work was supported by National Natural Science Foundation of China (21790052, T.L. and J.Z.; 51802220, X.C.) and the Natural Science Foundation of Tianjin City (19JCYBJC17300, X.C.).

## Author contributions

Y.L. and Z.Z. contributed equally to this work. X.C. designed the project, and X.C., T.L., and J.Z. supervised this project. Y.L. and Z.Z. fabricated the memory device and performed the electrical measurements. J.L. and Y.K. prepared the GDY film. F.W. and G.Z. performed the characterization. X.C. wrote the manuscript. All the authors discussed the results and commented on the manuscript.

## Competing interests

The authors declare no competing interests.
