## [Peer Review File · Nature Communications]

Title: Low-voltage ultrafast nonvolatile memory via direct charge injection through a threshold resistive-switching layerREVIEWER COMMENTS

Reviewer #1 (Remarks to the Author):

In essence, this work combines PCM with flash memories and uses direct charge transfer via conductive filaments in PCM to modulate the charge in the floating gate of flash memories.

The basic idea is interesting and the low operation voltage and high speed are noteworthy. However, I find the claims of fJ-level energy consumption misleading. The write operation seems to require 1uA level currents, this is typical of PCM, and with a 2V write voltage, this is more in the uJ level and is not competitive with other memory technologies or a rather long list of previously demonstrated various memory devices.

The choice of materials does not seem to be well-motivated, semiconductors are several layers thick and most of the materials are exfoliated by hand so this a bit dated fabrication approach and it is not clear how or even if this approach could be scaled to smaller devices or larger systems.

Reviewer #2 (Remarks to the Author):

Title: Low-voltage ultrafast nonvolatile memory via direct charge injection through a threshold resistive-switching layer

Author: Yuan Li, Zhi-Cheng Zhang, Jiaqiang Li, Xu-Dong Chen, Ya Kong, Fu-Dong Wang, Guo-Xin Zhang, Tong-Bu Lu, Jin Zhang

Summary:

The authors demonstrate a low voltage non-volatile memory by introducing a WSe₂/graphdiyne oxide/MoS₂ volatile threshold switching structure instead of an insulator between the control gate and floating gate. The threshold switching characteristic of this structure allows the charge to be injected in the MoS₂ floating gate from the WSe₂ control gate when a voltage is applied, and the charge is blocked in the floating gate when the voltage is removed. The switching time of the entire NVM cell which includes also a hBN blocking layer and a MoS₂ channel in addition to the resistive switching device is below 30 ns, while the retention is above 10 years. Moreover, the cell shows stable multi level storage.

Issues:

1. (page 1, rows 18-32). The abstract should be more concise. The cell structure should be mentioned here and the novelty of the structure.
2. (page 2, rows 33-42) In the general description of memory devices it is recommended to reference some studies related to other types of volatile switching devices such as [10.1038/s41598-017-08251-z, 10.1109/TED.2018.2862917], some of them also based on filamentary switching.
3. (page 4, row 98) Except for the thickness of the GDYO layer, the thicknesses of the other materials in

the structure are fixed and there is no comment regarding the motivation behind these values. For instance MoS₂ is 4.2 nm on bottom and 1.4 nm on top. Why?

4. (pages 16-18, rows 337-268) For the multibit storage in Fig. 6e I-t curves are only shown for 1000 s. Should we conclude that the retention time for each state is only 1000 s?

5. (page 19, rows 386-388) Authors claim several times that their structure can be used for 'next-generation memory technologies with ultrahigh speed, ultralong retention, ultrahigh capacity and ultralow energy consumption.' However, the current preparation method is not scalable. Which is the next step?

Typos/Language:

There are several language and editing errors such as:

(Manuscript) There are many dashes throughout the manuscript that make the article difficult to read such as: '20-nm', '20-ns', '±2-V/20-ns' etc. Please remove the dashes.

Reviewer #3 (Remarks to the Author):

The authors report a new memory device based on a van der Waals heterostructure. Short write/erase times (~20 ns), low write/erase voltages (~2 V) and long-retention times are demonstrated (up to 10 year). The work is timely, and the memory characteristics are competitive with previous reports of similar devices, of which there are many as summarized very nicely by the authors in Supplementary Table 1. Despite these many positive attributes, I have major reservations about this work:

1. The graphidyne oxide (GDYO) layer that is critical to the operation of the memory device is poorly characterized. This is especially important as GDYO is a poorly understood material as compared to the other materials used in the memory cell.

a) A more thorough XPS analysis, before and after oxidation, should be reported, including survey scans to show elemental composition. In particular, is there any silicon or other elements present during synthesis remaining in the final product? Higher resolution XPS scans for oxygen and trace elements detected by XPS survey should be reported.

b) FTIR measurements, before and after oxidation, should be reported to corroborate the structure shown in Supplementary Scheme 1, and provide further spectroscopic characterization of GDYO.

c) The pump wavelength that was used for Raman spectroscopy should be reported.

d) The UV-vis absorption spectrum of GDY and GDYO should be reported. Is there spectroscopic evidence of HOMO-LUMO gap opening upon oxidation?

2. The origin of resistive switching in graphene oxide thin films is a subject of active research (see ACS

Nano 2018, 12, 7, 7335–7342 for just one example, here with in-situ measurements). The authors should comment upon similarities and differences in resistive switching in GDYO versus graphene oxide.

3. The resistive switching behaviour of GDYO shown in Fig. 2 a and b is interesting. The authors interpret this response in terms of conductive filament formation, inspired by work with graphene oxide. What experimental evidence is there for filament formation? Have the authors studied the scaling of the resistive switching phenomenon systematically? For example, how do the set / reset voltages and currents scale with GDYO area, GDYO thickness, temperature, and other GDYO parameters such as oxygen content? This phenomenon is at the heart of the demonstrated device, but is poorly characterized.

A minor issue:

4. Vague terms such as "huge energy" and "lots of heat" are not appropriate in a scientific work. The abstract should be revised.

In summary, although the work is very interesting, I cannot recommend publication of this manuscript in Nature Communications in its present form. The characterization of the GDYO layer and its resistive switching behaviour is insufficient, especially given the central role played in achieving the reported results.

COMMENTS TO AUTHOR:

Reviewer #1 (Remarks to the Author):

In essence, this work combines PCM with flash memories and uses direct charge transfer via conductive filaments in PCM to modulate the charge in the floating gate of flash memories.

1. The basic idea is interesting and the low operation voltage and high speed are noteworthy. However, I find the claims of fJ-level energy consumption misleading. The write operation seems to require 1uA level currents, this is typical of PCM, and with a 2V write voltage, this is more in the uJ level and is not competitive with other memory technologies or a rather long list of previously demonstrated various memory devices.

Reply: We are grateful for the reviewer's positive comments and valuable suggestions for our work. We believe these kind comments can help us to improve the quality of our manuscript. We have revised the manuscript carefully according to these comments and the revised parts are marked by **blue** in the manuscript.

Energy consumption is an important parameter for the future electronic devices. For conventional PCMs, whose conductance states are switched between high-resistance amorphous phase and low-resistance crystalline phase via Joule heating, a relatively high current is always required for the RESET operation, leading to a high energy consumption. In contrast, a threshold switching layer (GDYO) is used to control the injection of electrons to the floating gate in our device. The resistance switching mechanism is the formation/rupture of the GDY conductive filaments (Supplementary Fig. 14). A sub-uA level current (approximately 0.2–0.3 uA, as shown in Figs. 2c,d) occurs while applying a 2 V voltage pulse, which is much smaller than that of conventional PCMs. Given the duration of the voltage pulse (Δt) is as short as 20 ns, the energy consumption can be calculated by the equation of $E = V_{CG} \times I_{CG} \times \Delta t = 2 \text{ V} \times 2.5 \times 10^{-7} \text{ A} \times 2 \times 10^{-8} \text{ s} = 10 \text{ fJ}$. This is a quite low energy consumption that exceeds most of the reported floating-gate memory devices.

We have added the calculation process to the manuscript for an easy understanding (Page 10, Lines 203-204).

2. The choice of materials does not seem to be well-motivated, semiconductors are several layers thick and most of the materials are exfoliated by hand so this a bit dated fabrication approach and it is not clear how or even if this approach could be scaled to smaller devices or larger systems.

Reply: We fully agree with the reviewer's comments that the motivation for the choice of the materials should be explained and the potential of the device for large-scale application should be demonstrated. In this work, we proposed an ultrafast nonvolatile memory device with a structure of MoS₂/hBN/MoS₂/GDYO/WSe₂. The most important innovation of this work is to propose a novel charge injection mechanism for floating gate memory that can achieve high-speed writing/erasing operations. To demonstrate the feasibility of this new mechanism, we designed this device structure and constructed the device with some mechanical exfoliated 2D materials including MoS₂, WSe₂ and hBN. In fact, besides the motivation as discussed below, one of the main reasons for the choice of these materials, including MoS₂, WSe₂, hBN and GDYO, is their ability for wafer-scale preparation¹⁻⁶, to meet the requirements for large-scale applications.

In our device, we used a bilayer MoS₂ as the channel since bilayer MoS₂ has better electrostatic control, smaller bandgap and higher mobility than monolayer, and bilayer MoS₂ is regarded as the sweet spot to balance device performance and power consumption⁶. MoS₂ and WSe₂ were chosen as the floating gate and control gate, respectively, since they can form an asymmetric energy band that facilitates the injection of electrons from control gate (WSe₂) to floating gate (MoS₂) and restricts the escape of electrons from floating gate. Here we used multilayer MoS₂ as the floating gate since multilayer MoS₂ has a larger density of states (DOS) than that of monolayer, which can induce better charge storage capability. On the other hand, considering the fact that the parasitic capacitive coupling between the floating gates of adjacent cells is enhanced obviously with the increases of floating gate thickness^{7,8}, the thickness of the floating gate should not be too thick. Thus here we used a 4.2 nm MoS₂ film as the floating gate to balance these two factors, which is comparable with the MoS₂ floating gate in previous work⁹.

Although bilayer and multilayer MoS₂ films are preferred to act as the channel and

floating gate, respectively, it is still essential to estimate the influence of MoS₂ thickness on device performance, especially for monolayer grown via chemical vapor deposition (CVD) due to its potential for large-scale applications. Here we investigated the performance of the memory devices using CVD-grown monolayer MoS₂ as channel and floating gate, respectively. As shown in Supplementary Figs. 31 and 32, these devices still possess nanosecond writing/erasing speed and excellent retention characteristics. We have also used CVD-grown monolayer WSe₂ as the control gate to construct the device, which demonstrated a comparable performance with the device based on mechanical exfoliated WSe₂ (Supplementary Fig. 33).

The GDYO layer acts as the threshold-switching layer, which is crucial for the ultrahigh operation speed and long retention time of the memory device. The influence of GDYO thickness on storage performance was investigated in Supplementary Figs. 28–30. For the device with a 3 nm GDYO layer, although ultrafast writing/erasing operations have been demonstrated, its retention characteristics suffer serious degradation (Supplementary Fig. 28). This is because thus a thin GDYO film has a large off-state conductance, leading to the escape of the injected electrons through the GDYO layer (Supplementary Fig. 30). On the contrary, as shown in Supplementary Fig. 29, the device with a 20 nm GDYO layer exhibits a long-term retention behavior, while V_{CG} pulses with larger amplitude and longer duration are required for the writing/erasing operations, since the switching of thus a thick GDYO film requires a larger SET voltage and longer switching time (Supplementary Fig. 30). Thus the optimal GDYO thickness evaluating by the operation speed/voltage and retention characteristics is ~10 nm for the memory device.

For the hBN layer, since electron tunneling mechanism was not utilized in our device, hBN was mainly used as a blocking layer to restrict the escape of charges from floating gate to the channel, and its thickness should be suitable. In fact, a too thin hBN film would cause the leakage of charges from floating gate to channel¹⁰, leading to a poor retention characteristic (Supplementary Fig. 34a). In turn, a too thick hBN is also not essential, especially considering that such a thick hBN film would share more voltage in the series circuit as illustrated in the inset in Supplementary Fig. 34b, leading to a

decrease of the effective voltage loaded on the GDYO layer. As a result, a degraded on/off ratio (10^4) was observed for the device with a 28 nm hBN blocking layer while applying ± 2 V, 20 ns V_{CG} pulses for writing and erasing operations. Thus here a ~ 10 nm hBN film was used as the blocking layer.

To demonstrate the potential of the device for large-scale applications, we have fabricated the device with CVD-grown large-area MoS₂ and WSe₂ films, and constructed a 5×5 device array as a demonstration (Supplementary Fig. 38). Noteworthy, here we used a 20 nm HfO₂ layer fabricated via atomic layer deposition (ALD) to replace hBN as the blocking layer, since high-quality large-area CVD-grown hBN film with thickness of ~ 10 nm was unavailable for us. Although multilayer MoS₂ is preferred for better performance, these devices based on CVD-grown TMDs still possess comparable performance with the device fabricated by mechanical exfoliated 2D materials. As shown in Supplementary Figs. 39 and 40, high-speed writing and erasing operations and long-term retention characteristics have also been demonstrated by all the 25 devices. The on- and off-state currents and the on/off ratio of these 25 devices were extracted (Supplementary Fig. 41). All of these 25 devices have an on/off ratio larger than 10^5 , with the largest one as 8×10^6 . Undoubtedly, these results demonstrate the potential of the ultrafast nonvolatile memory for large-scale applications based on CVD-grown 2D materials, especially considering the fact that the synthesis of large-area multilayer TMDs via CVD is going to mature in the near future⁶.

Thus far, the fabrication techniques for wafer-scale circuits based on 2D materials are gradually mature¹¹⁻¹³, and the use of CVD-grown 2D materials make it possible to fabricate device with smaller feature size. In this work, the devices were fabricated via photolithography in laboratory, and thus the feature size and the integration density of the devices were still limited. Here the channel length was approximately 2 μm . Nevertheless, we believe that nanoscale devices can be fabricated by using advanced industrial UV lithography and EBL techniques.

We have added these discussions and corresponding results to the manuscript (Pages 5,6, Lines 114-120; Pages 14,15, Lines 299-316; Pages 16,17, Lines 332-344) and Supplementary Information (Supplementary Figs. 28–34, 38–41).

Supplementary Fig. 28 | Storage performance of a memory device with a thin GDYO layer (3 nm). **a,b**, The successful ultrafast writing (**a**) and erasing (**b**) operations of the device by applying a 20 ns V_{CG} pulses with amplitude of ± 2 V, respectively. **c**, the retention performance of the device with a 3 nm GDYO layer. V_{ds} is fixed as 0.1 V for the readout operation.

Supplementary Fig. 29 | Storage performance of a memory device with a thick GDYO layer (20 nm). **a**, The switching of the channel currents by applying -5 V/ $+5$ V V_{CG} pulses with FWHM of 100 ns for the writing and erasing operations, respectively. **b**, the retention performance of the device with a 20 nm GDYO layer. V_{ds} is fixed as 0.1 V for the readout operation.

Supplementary Fig. 30| GDYO-based TS memories with different thicknesses of GDYO layer. **a**, The off-state currents of the devices with GDYO thickness of 3 nm (red) and 20 nm (blue), respectively. A fixed V_{ds} of 0.1 V was used to read the channel current. **b**, TS characteristics of devices with GDYO thickness of 3 nm (red), 10 nm (green) and 20 nm (blue), respectively. The red, green and blue circles mark the V_{SET} of these devices. **c**, Switching time of the TS device with a 20 nm GDYO layer. A -5 V/200 ns voltage pulse was used to set the device, and the read pulses were -0.1 V/100 ns. The off-state conductance of the device with a 3 nm GDYO layer is several orders of magnitude larger than that of the device with a 20 nm GDYO layer. For the memory device with a 3 nm GDYO layer, the GDYO TS layer cannot be completely turned off due to its large off-state conductance, leading to the escape of the injected electrons and thus poor retention characteristics as shown in Supplementary Fig. 28. On the other hand, with the increase of GDYO thickness, an increased SET voltage and longer switching time were observed, and thus V_{CG} pulses with larger amplitude and longer duration are required to switch the GDYO TS layer for the writing/erasing operations (Supplementary Fig. 29).

Supplementary Fig. 31 | Storage performance of a memory device using a CVD-grown monolayer MoS₂ as the channel. **a**, The switching of the channel currents by applying ± 2 V/20 ns V_{CG} pulses for the writing and erasing operations, respectively. **b**, the retention characteristics of the device at State-1 and State-0. V_{ds} is fixed as 0.1 V for the readout operation. Inset is the OM image of the device, and the CVD-grown MoS₂ is marked by the purple dotted triangle. Scale bar, 30 μ m.

Supplementary Fig. 32 | Storage performance of a memory device using a CVD-grown monolayer MoS₂ as the floating gate. **a**, The switching of the channel currents by applying ± 2 V/20 ns V_{CG} pulses for the writing and erasing operations, respectively. **b**, the retention characteristics of the device at State-1 and State-0. V_{ds} is fixed as 0.1 V for the readout operation. Inset is the OM image of the device, and the CVD-grown MoS₂ is marked by the green dotted triangle. Scale bar, 30 μ m.

Supplementary Fig. 33 | Storage performance of a memory device using a CVD-grown monolayer WSe₂ as the control gate. **a**, The switching of the channel currents by applying ± 2 V/20 ns V_{CG} pulses for the writing and erasing operations, respectively. **b**, the retention characteristics of the device at State-1 and State-0. V_{ds} is fixed as 0.1 V for the readout operation. Inset is the OM image of the device, and the CVD-grown WSe₂ is marked by the red dotted triangle. Scale bar, 30 μ m.

Supplementary Fig. 34 | Storage performance of devices with a hBN blocking layer of 3 nm (**a**) and 28 nm (**b**), respectively. The thickness of hBN blocking layer have a significant influence on device performance. For the device with a 3 nm hBN film, the blocking layer is too thin to restrict the charges in the floating gate, leading to the leakage of charges from floating gate to channel. Thus the device exhibits poor retention characteristics after the writing and erasing operations. The inset in **b** illustrates the equivalent circuit of the memory device while applying a V_{CG} pulse, in which the hBN and GDYO are connected in series. Considering that a thicker hBN film has a larger resistance, which will share more voltage. For the device with a 28 nm hBN layer, the

effective voltage loaded on the GDYO TS layer is reduced in comparison with that of the device with a 10 nm hBN film, which affects the writing and erasing operations with a degraded on/off ratio (10^4). Thus the thickness of hBN blocking layer should also be carefully chosen, and a 10 nm thick hBN film is suitable for our memory device.

Supplementary Fig. 38 | 5×5 device array fabricated by CVD-grown large-area MoS₂ and WSe₂ films. Scale bars, 50 μm (left) and 10 μm (right). Monolayer CVD-grown WSe₂ was first patterned into rectangle array as the control gate of the devices, followed by the deposition of Cr/Au as the gate electrodes connecting the WSe₂. Then the prepatterned GDYO and MoS₂ (CVD-grown) layers were stacked on top in turn, acting as the TS layer and floating gate, respectively. Noteworthy, a 20 nm thick HfO₂ layer instead of hBN film fabricated via atomic layer deposition (ALD) was used as the blocking layer between the floating gate and channel. Finally, prepatterned CVD-grown MoS₂ was stacked on top of the HfO₂, acting as the channel, and the source and drain electrodes (Cr/Au) were deposited.

Supplementary Fig. 39 Ultrafast writing and erasing operations of 25 memory devices based on CVD-grown MoS₂ and WSe₂ as illustrated in Supplementary Fig. 38. A -2 V, 20 ns and a $+2$ V, 20 ns V_{CG} pulses were used for the writing and erasing operations, respectively, and the readout voltage was fixed at 0.1 V.

Supplementary Fig. 40 Retention characteristics of 25 memory devices based on CVD-grown MoS₂ and WSe₂ as illustrated in Supplementary Fig. 38. The readout voltage was fixed as 0.1 V.

Supplementary Fig. 41 | The currents of the 25 devices at State-0 and State-1 and the on/off ratio extracted from Supplementary Fig. 40.

Reviewer #2 (Remarks to the Author):

Title: Low-voltage ultrafast nonvolatile memory via direct charge injection through a threshold resistive-switching layer

Author: Yuan Li, Zhi-Cheng Zhang, Jiaqiang Li, Xu-Dong Chen, Ya Kong, Fu-Dong Wang, Guo-Xin Zhang, Tong-Bu Lu, Jin Zhang

Summary:

The authors demonstrate a low voltage non-volatile memory by introducing a WSe₂/graphdiyne oxide/MoS₂ volatile threshold switching structure instead of an insulator between the control gate and floating gate. The threshold switching characteristic of this structure allows the charge to be injected in the MoS₂ floating gate from the WSe₂ control gate when a voltage is applied, and the charge is blocked in the floating gate when the voltage is removed. The switching time of the entire NVM cell which includes also a hBN blocking layer and a MoS₂ channel in addition to the resistive switching device is below 30 ns, while the retention is above 10 years. Moreover, the cell shows stable multi level storage.

Reply: We are grateful for the reviewer’s positive comments and valuable suggestions for our work. We believe these kind comments can help us to improve the quality of our manuscript. We have revised the manuscript carefully according to these comments and the revised parts are marked by **blue** in the manuscript.

Issues:

1. (page 1, rows 18-32). The abstract should be more concise. The cell structure should be mentioned here and the novelty of the structure.

Reply: We fully agree with the reviewer's comment, and we have revised the abstract following the reviewer's suggestion.

2. (page 2, rows 33-42) In the general description of memory devices it is recommended to reference some studies related to other types of volatile switching devices such as [10.1038/s41598-017-08251-z, 10.1109/TED.2018.2862917], some of them also based on filamentary switching.

Reply: Many thanks for the reviewer's kind reminder. We have revised our Introduction and referenced some studies on volatile switching devices following the reviewer's suggestion (Refs. 12, 13).

3. (page 4, row 98) Except for the thickness of the GDYO layer, the thicknesses of the other materials in the structure are fixed and there is no comment regarding the motivation behind these values. For instance MoS₂ is 4.2 nm on bottom and 1.4 nm on top. Why?

Reply: We are grateful for the reviewer's valuable comments. In this work, the volatile threshold switching layer, GDYO, plays a key role in our ultrafast nonvolatile memory, and thus we systematically investigated the influence of the thickness of the GDYO layer on the device performance. Of course, just as the reviewer said, the motivation and influence of the thickness of other materials should also be discussed.

For the channel, here a bilayer MoS₂ film was used in our device, since bilayer MoS₂ has better electrostatic control, smaller bandgap and higher mobility than monolayer, and bilayer MoS₂ is regarded as the sweet spot to balance device performance and power consumption⁶. On the other hand, thick MoS₂ is preferred as the floating gate since multilayer MoS₂ can store more charges due to its larger density of states. However, reducing the floating gate thickness is regarded as an effective way to suppress the parasitic capacitive coupling between the floating gates of adjacent cells^{7,8}.

Thus we chose a 4 nm thick MoS₂ as the floating gate to balance these two factors.

Although bilayer and multilayer MoS₂ films are preferred to act as the channel and floating gate, respectively, it is still essential to estimate the influence of MoS₂ thickness to device performance, especially for monolayer grown via chemical vapor deposition (CVD) due to its potential for large-scale applications. Here we investigated the performance of the memory devices using CVD-grown monolayer MoS₂ as channel and floating gate, respectively. As shown in Supplementary Figs. 31 and 32, these devices still possess nanosecond writing/erasing speed and excellent retention characteristics. We have also used CVD-grown monolayer WSe₂ as the control gate to construct the device, which demonstrated a comparable performance with the device based on mechanical exfoliated WSe₂ (Supplementary Fig. 33).

For the dielectric blocking layer, its thickness should be suitable. In fact, a too thin hBN film would cause the leakage of charges from floating gate to channel¹⁰, leading to a poor retention characteristic (Supplementary Fig. 34a). In turn, a too thick hBN is also not essential, especially considering that such a thick hBN film would share more voltage in the series circuit as illustrated in the inset in Supplementary Fig. 34b, leading to the decrease of the effective voltage loaded on the GDYO layer. As a result, a degraded on/off ratio (10^4) was observed for the device with a 28 nm hBN blocking layer while applying ± 2 V, 20 ns V_{CG} pulses for writing and erasing operations. Thus here a ~ 10 nm hBN film was used as the blocking layer.

We have added these discussions and corresponding results to the manuscript (Pages 5,6, Lines 114-120; Page 15, Lines 309-316) and Supplementary Information (Supplementary Figs. 31–34).

Supplementary Fig. 31 | Storage performance of a memory device using a CVD-grown monolayer MoS₂ as the channel. **a**, The switching of the channel currents by applying ± 2 V/20 ns V_{CG} pulses for the writing and erasing operations, respectively. **b**, the retention characteristics of the device at State-1 and State-0. V_{ds} is fixed as 0.1 V for the readout operation. Inset is the OM image of the device, and the CVD-grown MoS₂ is marked by the purple dotted triangle. Scale bar, 30 μ m.

Supplementary Fig. 32 | Storage performance of a memory device using a CVD-grown monolayer MoS₂ as the floating gate. **a**, The switching of the channel currents by applying ± 2 V/20 ns V_{CG} pulses for the writing and erasing operations, respectively. **b**, the retention characteristics of the device at State-1 and State-0. V_{ds} is fixed as 0.1 V for the readout operation. Inset is the OM image of the device, and the CVD-grown MoS₂ is marked by the green dotted triangle. Scale bar, 30 μ m.

Supplementary Fig. 33 | Storage performance of a memory device using a CVD-grown monolayer WSe₂ as the control gate. **a**, The switching of the channel currents by applying ± 2 V/20 ns V_{CG} pulses for the writing and erasing operations, respectively. **b**, the retention characteristics of the device at State-1 and State-0. V_{ds} is fixed as 0.1 V for the readout operation. Inset is the OM image of the device, and the CVD-grown WSe₂ is marked by the red dotted triangle. Scale bar, 30 μ m.

Supplementary Fig. 34 | Storage performance of devices with a hBN blocking layer of 3 nm (**a**) and 28 nm (**b**), respectively. The thickness of hBN blocking layer have a significant influence on device performance. For the device with a 3 nm hBN film, the blocking layer is too thin to restrict the charges in the floating gate, leading to the leakage of charges from floating gate to channel. Thus the device exhibits poor retention characteristics after the writing and erasing operations. The inset in **b** illustrates the equivalent circuit of the memory device while applying a V_{CG} pulse, in which the hBN and GDYO are connected in series. Considering that a thicker hBN film has a larger resistance, which will share more voltage. For the device with a 28 nm hBN layer, the

effective voltage loaded on the GDYO TS layer is reduced in comparison with that of the device with a 10 nm hBN film, which affects the writing and erasing operations with a degraded on/off ratio (10^4). Thus the thickness of hBN blocking layer should also be carefully chosen, and a 10 nm thick hBN film is suitable for our memory device.

4. (pages 16-18, rows 337-268) For the multibit storage in Fig. 5e I - t curves are only shown for 1000 s. Should we conclude that the retention time for each state is only 1000 s?

Reply: Many thanks for the reviewer's kind reminder. To demonstrate the retention characteristics of the memory device in a longer timescale, we have measured the I - t curves of the device at different storage levels for 10^4 s. As shown in Fig. 5e, the device still exhibits distinct storage levels even after such a long time.

We have added this result to Fig. 5e to replace the initial one.

Fig. 5e, I - t curves of the memory device at 8 distinct conductive levels for 10^4 s.

5. (page 19, rows 386-388) Authors claim several times that their structure can be used for 'next-generation memory technologies with ultrahigh speed, ultralong retention, ultrahigh capacity and ultralow energy consumption.' However, the current preparation method is not scalable. Which is the next step?

Reply: We fully agree with the reviewer's valuable comments. In this work, we used

some mechanical exfoliated 2D materials, such as MoS₂, WSe₂ and hBN, to construct the device, which are not feasible for large-scale fabrication. In fact, when we chose the materials for the device, we have considered the potential of the device for large-scale applications. One of the main reasons for us to choose these materials, including MoS₂, WSe₂, hBN and GDYO, is their ability for wafer-scale preparation¹⁻⁶, to meet the requirements for large-scale applications.

To demonstrate the potential of the device for large-scale applications, we have fabricated the device with CVD-grown large-area MoS₂ and WSe₂ films, and constructed a 5 × 5 device array as a demonstration (Supplementary Fig. 38). Noteworthy, here we used a 20 nm HfO₂ layer fabricated via atomic layer deposition (ALD) to replace hBN as the blocking layer, since high-quality large-area CVD-grown hBN film with thickness of ~10 nm was unavailable for us. Although multilayer MoS₂ is preferred for better performance, these devices based on CVD-grown TMDs still possess comparable performance with the device fabricated by mechanical exfoliated 2D materials. As shown in Supplementary Figs. 39 and 40, high-speed writing and erasing operations and long-term retention characteristics have also been demonstrated by all the 25 devices. The on- and off-state currents and the on/off ratio of these 25 devices were extracted (Supplementary Fig. 41). All of these 25 devices have an on/off ratio larger than 10⁵, with the largest one as 8 × 10⁶. Undoubtedly, these results demonstrate the potential of the ultrafast nonvolatile memory for large-scale applications based on CVD-grown 2D materials.

We have added these discussions and corresponding results to the manuscript (Pages 16,17, Lines 332-344) and Supplementary Information (Supplementary Figs. 38–41).

Supplementary Fig. 38 | 5×5 device array fabricated by CVD-grown large-area MoS₂ and WSe₂ films. Scale bars, 50 μm (left) and 10 μm (right). Monolayer CVD-grown WSe₂ was first patterned into rectangle array as the control gate of the devices, followed by the deposition of Cr/Au as the gate electrodes connecting the WSe₂. Then the prepatterned GDYO and MoS₂ (CVD-grown) layers were stacked on top in turn, acting as the TS layer and floating gate, respectively. Noteworthily, a 20 nm thick HfO₂ layer instead of hBN film fabricated via atomic layer deposition (ALD) was used as the blocking layer between the floating gate and channel. Finally, prepatterned CVD-grown MoS₂ was stacked on top of the HfO₂, acting as the channel, and the source and drain electrodes (Cr/Au) were deposited.

Supplementary Fig. 39 Ultrafast writing and erasing operations of 25 memory devices based on CVD-grown MoS₂ and WSe₂ as illustrated in Supplementary Fig. 38. A -2 V, 20 ns and a $+2$ V, 20 ns V_{CG} pulses were used for the writing and erasing operations, respectively, and the readout voltage was fixed at 0.1 V.

Supplementary Fig. 40 Retention characteristics of 25 memory devices based on CVD-grown MoS₂ and WSe₂ as illustrated in Supplementary Fig. 38. The readout voltage was fixed as 0.1 V.

Supplementary Fig. 41 | The currents of the 25 devices at State-0 and State-1 and the on/off ratio extracted from Supplementary Fig. 40.

Typos/Language:

There are several language and editing errors such as:

(Manuscript) There are many dashes throughout the manuscript that make the article difficult to read such as: ‘20-nm’, ‘20-ns’, ‘ ± 2 -V/20-ns’ etc. Please remove the dashes.

Reply: Many thanks for the reviewer’s kind reminder, and we have revised these errors in our manuscript and Supplementary Information.

Reviewer #3 (Remarks to the Author):

The authors report a new memory device based on a van der Waals heterostructure. Short write/erase times (~ 20 ns), low write/erase voltages (~ 2 V) and long-retention times are demonstrated (up to 10 year). The work is timely, and the memory characteristics are competitive with previous reports of similar devices, of which there are many as summarized very nicely by the authors in Supplementary Table 1. Despite these many positive attributes, I have major reservations about this work:

Reply: We are grateful for the reviewer’s positive comments and valuable suggestions for our work. We believe these kind comments can help us to improve the quality of our manuscript. We have revised the manuscript carefully according to these comments and the revised parts are marked by **blue** in the manuscript.

1. The graphdiyne oxide (GDYO) layer that is critical to the operation of the memory device is poorly characterized. This is especially important as GDYO is a poorly understood material as compared to the other materials used in the memory cell.

a) A more thorough XPS analysis, before and after oxidation, should be reported, including survey scans to show elemental composition. In particular, is there any silicon or other elements present during synthesis remaining in the final product? Higher resolution XPS scans for oxygen and trace elements detected by XPS survey should be reported.

Reply: We fully agree with the reviewer's comments. To analyze the elemental composition of the GDYO, the XPS of the GDY film before and after oxidation have been measured on both SiO₂/Si and sapphire substrate. Supplementary Figs. 2a,b depict the survey scans of the GDY and GDYO films on SiO₂/Si substrate. For the GDY film before oxidation, its O 1s peak can be deconvoluted into three subpeaks corresponding to Si–O, C–O and C=O bonds (Supplementary Fig. 2c), and the Si–O bond from SiO₂ substrate is dominant. After UV-ozone treatment, a new subpeak corresponding to O=C–OH bond appears, and the proportion of C=O bonds is significantly increased (Supplementary Fig. 2d). Similarly, an obvious increase of subpeaks for the O=C–OH and C=O bonds are also observed in the C 1s peak (Supplementary Figs. 2e,f). These results demonstrate the oxidation of GDY film via UV-ozone treatment.

To analyze the trace elements such as Si and Cu induced during the synthesis of GDY, the XPS of GDY film before and after oxidation were also measured on sapphire substrate. Supplementary Figs. 3a–c depict the survey scans of the GDY and GDYO films on sapphire, using a bare sapphire as reference. Supplementary Figs. 3d–f present the high-resolution Si 2p peaks of bare sapphire, GDY and GDYO films on sapphire, respectively. Noteworthy, a weak peak of Si 2p (98.3 eV) was observed even on bare sapphire which might originate from impurities in sapphire. For the GDY and GDYO films, a new subpeak appears at 102.4 eV, indicating that trace Si was induced to the GDY from the HEB-TMS monomers during the synthesis process. Similarly, the appearance of a weak Cu 2p peak demonstrates that GDY and GDYO films contain trace Cu induced from the Cu foil during the synthesis process (Supplementary Figs.

3g-i).

We have added these results to the Supplementary Information (Supplementary Page 3; Supplementary Figs. 2, 3).

Supplementary Fig. 2 | XPS of GDY and GDYO films on SiO₂/Si substrate. **a,b**, Survey scans of the GDY (**a**) and GDYO films (**b**) on SiO₂/Si substrate. **c,d**, High resolution XPS of O 1s peak for GDY (**c**) and GDYO films (**d**), respectively. **e,f**, High resolution XPS of C 1s peak for GDY (**e**) and GDYO films (**f**), respectively.

Supplementary Fig. 3 | XPS of GDY and GDYO films on sapphire substrate. **a–c**, Survey scans of bare sapphire (**a**), GDY (**b**) and GDYO films (**c**) on sapphire substrate, respectively. **d–f**, High resolution XPS of Si 2p peak for bare sapphire (**d**), GDY (**e**) and GDYO films (**f**) on sapphire substrate, respectively. **g–i**, High resolution XPS of Cu 2p peak for bare sapphire (**g**), GDY (**h**) and GDYO films (**i**) on sapphire substrate, respectively.

b) FTIR measurements, before and after oxidation, should be reported to corroborate the structure shown in Supplementary Scheme 1, and provide further spectroscopic characterization of GDYO.

Reply: We are grateful for the reviewer's valuable comment. The FTIR of GDY and GDYO films were measured (Supplementary Fig. 5). For the GDY film, the bands located at 1627 cm^{-1} is attributed to the skeletal vibration of benzene ring, while the wide band at 2107 cm^{-1} is due to the stretching vibration of $\text{C}\equiv\text{C}$ bond¹⁴. In comparison, the GDYO film after UV-ozone treatment has an enhanced band at 1103 cm^{-1} and 1727 cm^{-1} , which are ascribed to the stretching vibration of C–O and C=O bonds^{15,16}, and the

band corresponding to $C\equiv C$ almost disappears. The band located around 619 cm^{-1} is attributed to the bending vibration of $O-H$ ¹⁵. These results suggest that a large amount of oxygen-containing groups are bonded to GDY by UV-ozone treatment, and the *sp*-hybrid carbon atoms ($C\equiv C$) are the main oxidation sites.

We have added these results to the Supplementary Information (Supplementary Page 4; Supplementary Fig. 5).

Supplementary Fig. 5 | FTIR spectra of GDY (blue) and GDYO (red).

c) The pump wavelength that was used for Raman spectroscopy should be reported.

Reply: Many thanks for the reviewer's kind reminder. Raman spectroscope (Horiba Evolution) equipped with a 532 nm pump laser was used for the Raman spectra measurement. We have added this description to the Methods section in manuscript (Page 21, Lines 424-425).

d) The UV-vis absorption spectrum of GDY and GDYO should be reported. Is there spectroscopic evidence of HOMO-LUMO gap opening upon oxidation?

Reply: We are grateful for the reviewer's valuable comments. The UV-vis absorption spectra of GDY and GDYO films were measured (Supplementary Fig. 6). The band gaps of GDY and GDYO films were evaluated by the Tauc plots as shown in Supplementary Figs. 6b,c, which were 2.40 eV and 4.14 eV, respectively. Thus the band

gap of GDYO is enlarged in comparison with that of GDY. Moreover, the UPS of GDY and GDYO were also measured. As shown in Supplementary Fig. 7, the Fermi level of GDY and GDYO are calculated as 5.01 eV and 4.84 eV, respectively, and the valence bands of GDY and GDYO are 6.40 eV and 7.08 eV, respectively. Combining the optical band gaps obtained from the absorption spectra, which are 2.40 eV (GDY) and 4.14 eV (GDYO), the conduction bands of GDY and GDYO are 4.00 eV and 2.94 eV, respectively. Thus we can obtain the band diagrams of GDY and GDYO as illustrated in Supplementary Fig. 7e. These spectroscopic evidences can demonstrate the HOMO-LUMO gap opening of GDY upon oxidation.

We have added these results to the Supplementary Information (Supplementary Page 4; Supplementary Figs. 6, 7).

Supplementary Fig. 6 | UV-vis-NIR absorbance spectra (a) and corresponding Tauc plots (b,c) of GDY and GDYO films. The band gaps of GDY and GDYO extracted from the Tauc plots are 2.40 eV and 4.14 eV, respectively.

Supplementary Fig. 7 | Band structure of GDY and GDYO measured by UPS spectra. **a–d**, UPS spectra of GDY (**a,b**) and GDYO (**c,d**) films, respectively. **e**, Band diagrams of GDY and GDYO. The Fermi level can be calculated using the equation $E_F = h\nu - E_{\text{onset}}$, where $h\nu$ is the incident photon energy (21.22 eV), and E_{onset} is the onset level related to the secondary electrons¹⁷. Thus the Fermi levels of GDY and GDYO are 5.01 eV and 4.84 eV, respectively. The cutoff of the lowest binding energy (as illustrated in **b** and **d**) indicates the difference between the energy of Fermi level and valence band maximum ($E_V - E_F$)¹⁷. and thus the valence bands of GDY and GDYO are calculated as 6.40 eV and 7.08 eV, respectively. Combing the optical band gaps obtained from the Tauc plots as shown in Supplementary Fig. 6, which are 2.40 eV (GDY) and 4.14 eV (GDYO), the conduction bands of GDY and GDYO are 4.00 eV and 2.94 eV, respectively. Thus we can obtain the band diagrams of GDY and GDYO as illustrated in **e**.

2. The origin of resistive switching in graphene oxide thin films is a subject of active research (see ACS Nano 2018, 12, 7, 7335–7342 for just one example, here with in-situ measurements). The authors should comment upon similarities and differences in resistive switching in GDYO versus graphene oxide.

Reply: Many thanks for the reviewer’s valuable suggestion. Memristors using graphene oxide (GO) as the active layer have been widely investigated, and the resistive switching of the device is mainly induced by the migration of oxygen-containing groups

driven by external electric field^{18,19}. Similarly, in this work, the switching of the GDYO conductance is also attributed to the oxygen-containing group migration, which can form GDY conductive filaments.

The main difference in the resistive switching behaviors between GO- and GDYO-based devices is their volatility. The GO-based memristors always exhibit nonvolatile resistive switching or even write-once-read-many characteristics¹⁸⁻²¹, while the GDYO-based device in this work presents a volatile threshold switching behavior. This is mainly attributed to the difference in the structure of graphene and GDY. For the GDYO-based memristor, when a bias voltage is applied, the oxygen-containing groups migrate driven by the electric field, forming GDY conductive filaments. As a result, the device switches from HRS to LRS. When the bias voltage is removed, the ordered porous structure of GDY (as illustrated in Supplementary Fig. 14d) enables the diffusion of oxygen-containing groups back to oxygen-deficient regions, driven by the concentration difference. Thus the device switches back to initial HRS spontaneously, demonstrating a volatile threshold switching behavior. In contrast, for the GO-based memristor, the dense honeycomb structure of graphene leads to a large diffusion barrier for oxygen-containing groups, and the migrated oxygen-containing groups are hard to diffuse back to their initial regions spontaneously, leading to a nonvolatile resistive switching characteristic. For our memory device, the connection between the control gate and floating gate should be ruptured to restrict the escape of injected electrons from the floating gate, and thus the GDYO layer with volatile threshold switching behavior is optimal.

We have added these discussions to the manuscript (Page 7, Lines 142-160) and Supplementary Information (Supplementary Fig. 14).

Supplementary Fig. 14 | Schematic illustration of the mechanism for the device conductance switching. **a**, GDYO film containing numerous oxygen-containing groups at initial HRS. **b**, By applying a bias voltage, the oxygen-containing groups migrate, forming GDY conductive filaments in the oxygen-deficient regions. As a result, the device switches from HRS to LRS. **c**, After removing the bias voltage, the concentration difference of oxygen-containing groups drive the diffusion of oxygen-containing groups back to oxygen-deficient regions, rupturing GDY conductance filaments. Thus the device returns back to its initial HRS spontaneously. **d**, Illustration of the ordered porous structure of GDY that enables the diffusion of oxygen-containing groups through the GDY layer.

3. The resistive switching behavior of GDYO shown in Fig. 2a and b is interesting. The authors interpret this response in terms of conductive filament formation, inspired by work with graphene oxide. What experimental evidence is there for filament formation? Have the authors studied the scaling of the resistive switching phenomenon systematically? For example, how do the set/reset voltages and currents scale with GDYO area, GDYO thickness, temperature, and other GDYO parameters such as

oxygen content? This phenomenon is at the heart of the demonstrated device, but is poorly characterized.

Reply: We are grateful for the reviewer's valuable comments. The XPS and Raman spectra as shown in Supplementary Figs. 2 and 4 demonstrate that the oxygen-containing groups mainly bind to the *sp*-hybrid carbon atoms ($C\equiv C$). The Raman Y' band at around 2191 cm^{-1} corresponding to $C\equiv C$ bond has an obvious difference between GDY and GDYO. Thus we can use Raman mapping to monitor the formation of GDY filaments while applying a bias voltage. The Raman Y' band map was first measured before applying bias voltage, and the result as shown in Supplementary Fig. 15a is quite weak. While applying a 2 V bias voltage, the intensity of Raman Y' band map has a significant enhancement (Supplementary Fig. 15b), demonstrating the formation of GDY as the conductive filaments (CFs). After removing the bias voltage, the intensity of Raman Y' band map returns to its initial level (Supplementary Fig. 15c), which indicates that the GDY filaments are disappeared. Furthermore, we also performed the conductive AFM to demonstrate the conductance switching of the GDYO layer. A small bias voltage (0.4 V) smaller than V_{SET} was first applied, and the measured currents were only several picoamperes (Supplementary Fig. 16a). While a bias voltage (1.0 V) exceeding V_{SET} was applied, the GDYO film switched from HRS to LRS with current of tens of nanoamperes (Supplementary Fig. 16b). After removing the large bias, a small bias voltage of 0.4 V was applied again, and the measured currents returned to picoampere level (Supplementary Fig. 16c). These results are consistent with the GDY filament formation and rupture as demonstrated in Supplementary Fig. 15. The Raman Y' band maps and conductive AFM measurements provide strong experimental evidence to demonstrate that the resistance switching of GDYO is attributed to the formation/rupture of GDY CFs.

Following the reviewer's comments, we have systematically studied the resistive switching characteristics of GDYO with different areas, thickness, temperatures and oxygen content. Supplementary Fig. 17 shows the *I-V* curves of the devices with GDYO areas scaling from $50 \times 50\ \mu\text{m}^2$ to $2 \times 2\ \mu\text{m}^2$. These devices possess similar SET and RESET voltages due to their localized filamentary switching nature²², while their on-

state currents have a proportional decrease with device size decreasing. Supplementary Fig. 18 depicts the resistive switching characteristics of the devices with GDYO thickness of 5 nm, 10 nm, 15 nm and 20 nm. With the increase of GDYO thickness, the SET/RESET voltages have an obvious increase. Noteworthy, the off-state current of the device with a 5 nm thick GDYO is approximately two orders of magnitude larger than that of other devices, which might lead to the leakage of electrons from the floating gate and degrade the storage performance of the memory device. Thus the optimal GDYO thickness evaluating by the SET voltage and HRS current is ~10 nm for the memory device. The temperature-dependent resistive switching characteristics of the device is shown in Supplementary Fig. 19. An obviously decreased SET voltage was observed with temperature increasing, since higher temperature can intensify the movement of oxygen-containing groups and accelerate the formation of GDY CFs. However, higher temperature also leads to a larger HRS current, which would sacrifice the retention performance of the memory device as demonstrated in Supplementary Fig. 43. GDYO films with different oxygen content were prepared by controlling the time of UV-ozone treatment. Supplementary Fig. 20 shows the resistive switching characteristics of the GDYO-based devices with oxidation treatment time of 30 s–120 s. With the decrease of oxidation time (oxygen content), the on/off of the device has a serious degradation due to the significant increase of the HRS current, and the SET/RESET voltages also show an obvious decrease. Noteworthy, for the device with a 30 s oxidation time, it cannot switch back to HRS after voltage sweeping due to the low concentration of oxygen-containing groups in this device.

We have added these discussions and corresponding results to the manuscript (Page 7, Lines 152-154, 161; Page 9, Line 177) and Supplementary Information (Supplementary Figs. 15–20).

Supplementary Fig. 15 Raman Y' band maps of GDYO film at pristine state (a), applying a 2 V bias voltage (b) and after removing the bias voltage (c). Scale bars, 2 μm . d–f, Raman profiles of the yellow point in a–c, respectively. Since the main oxidation sites of GDY are the *sp*-hybrid carbon atoms, the Raman Y' band corresponding to C \equiv C bond has an obvious difference between GDY and GDYO. For the pristine GDYO before applying bias voltage, its Raman Y' band is quite weak. While applying a 2 V bias voltage, the intensity of Raman Y' band has a significant enhancement, demonstrating the formation of GDY as the conductive filaments. After removing the bias voltage, the intensity of Raman Y' band returns to its initial level, which indicates that the GDY filaments are disappeared.

Supplementary Fig. 16 | Conductive AFM images of GDYO film measured with different bias voltages. **a**, While a bias voltage (0.4 V) smaller than V_{SET} (approximately 0.7 V) was applied, the GDYO film remains at HRS with a picoampere current level. **b**, While a bias voltage (1.0 V) exceeding V_{SET} was applied, the GDYO film switched to LRS with a nanoampere current level, demonstrating the formation of GDY CFs. **c**, Followed by the measurement in **b**, a bias voltage of 0.4 V was applied again, and the GDYO film returned to its HRS with a picoampere current level, which indicates the rupture of GDY CFs after removing the bias voltage. Noteworthy, the picoampere current level in **a** and **c** is the measuring limit of the instrument.

Supplementary Fig. 17 Resistive switching characteristics of the TS devices with different GDYO areas. **a**, I - V curves of the devices with GDYO areas scaling from $50 \times 50 \mu\text{m}^2$ to $2 \times 2 \mu\text{m}^2$. **b,c**, Distribution of the SET/RESET voltages (**b**) and HRS/LRS currents (**c**) of the devices with different sizes. These devices possess similar SET and RESET voltages due to their localized filamentary switching nature²², while the LRS currents have a proportional increase with device size increasing.

Supplementary Fig. 18 Resistive switching characteristics of the TS devices with different GDYO thickness. **a**, I - V curves of the devices with GDYO thickness of 5 nm, 10 nm, 15 nm and 20 nm. **b,c**, Distribution of the SET/RESET voltages (**b**) and HRS/LRS currents (**c**) of the devices with different thickness. With the increase of GDYO thickness, the SET/RESET voltages of the device have an obvious increase, while their LRS currents decrease correspondingly. For the device with a 20 nm thick GDYO, its SET voltage exceeds -1 V (approximately -1.3 V as shown in Supplementary Fig. 30), and thus it maintained at HRS during the voltage sweeping from 0 V to -1 V and back to 0 V. Noteworthy, for the device with a 5 nm thick GDYO,

its HRS current is approximately two orders of magnitude larger than that of other devices, which might lead to the leakage of electrons from floating gate and degrade the storage performance of the memory device. Thus the optimal GDYO thickness evaluating by the SET voltage and HRS current is ~ 10 nm for the memory device as demonstrated in Supplementary Figs. 28 and 29.

Supplementary Fig. 19 | Temperature-dependent resistive switching characteristics of the TS device. **a**, I - V curves of the devices measured at temperature ranging from 240 K to 340 K. **b,c**, Distribution of the SET/RESET voltages (**b**) and HRS/LRS currents (**c**) of the devices as a function of temperature. An obviously decreased SET voltage was observed with temperature increasing, since higher temperature can intensify the movement of oxygen-containing groups and accelerate the formation of GDY CFs. As well, higher temperature leads to a larger HRS current, which would sacrifice the retention performance of the memory device as demonstrated in Supplementary Fig. 43.

Supplementary Fig. 20 | Resistive switching characteristics of the TS devices with

different oxidation treatment time. **a**, *I-V* curves of the devices with oxidation time ranging from 30 s to 120 s. **b,c**, Distribution of the SET/RESET voltages (**b**) and HRS/LRS currents (**c**) of the devices as a function of oxidation time. The oxygen content in GDYO increases while prolonging UV-ozone treatment time. With the decrease of oxidation time (oxygen content), the on/off ratio of the device has a serious degradation due to the significant increase of the HRS current, and the SET/RESET voltages also show an obvious decrease. Noteworthily, for the device with a 30 s oxidation time, it cannot switch back to HRS after voltage sweeping due to the low concentration of oxygen-containing groups in this device.

A minor issue:

4. Vague terms such as "huge energy" and "lots of heat" are not appropriate in a scientific work. The abstract should be revised.

Reply: We are grateful for the reviewer's kind reminder, and we have revised the abstract following the reviewer's suggestion.

In summary, although the work is very interesting, I cannot recommend publication of this manuscript in Nature Communications in its present form. The characterization of the GDYO layer and its resistive switching behavior is insufficient, especially given the central role played in achieving the reported results.

References

1. Yang, P. F. et al. Batch production of 6-inch uniform monolayer molybdenum disulfide catalyzed by sodium in glass. *Nat. Commun.* **9**, 979 (2018).
2. Wang, L. et al. Epitaxial growth of a 100-square-centimetre single-crystal hexagonal boron nitride monolayer on copper. *Nature* **570**, 91-95 (2019).
3. Li, J. et al. Synthesis of wafer-scale ultrathin graphdiyne for flexible optoelectronic memory with over 256 storage levels. *Chem* **7**, 1284-1296 (2021).
4. Li, J. et al. General synthesis of two-dimensional van der Waals heterostructure arrays. *Nature* **579**, 368-374 (2020).
5. Wang, Y. et al. Wafer-scale synthesis of monolayer WSe₂: A multi-functional photocatalyst for efficient overall pure water splitting. *Nano Energy* **51**, 54-60 (2018).

6. Liu, L. et al. Uniform nucleation and epitaxy of bilayer molybdenum disulfide on sapphire. *Nature* **605**, 69-75 (2022).
7. Jae-Duk, L., Sung-Hoi, H. & Jung-Dal, C. Effects of floating-gate interference on nand flash memory cell operation. *IEEE Electr. Device L.* **23**, 264-266 (2002).
8. Misra, A. et al. Multilayer graphene as charge storage layer in floating gate flash memory. *2012 4th IEEE International Memory Workshop*. 1-4.
9. Sup Choi, M. et al. Controlled charge trapping by molybdenum disulphide and graphene in ultrathin heterostructured memory devices. *Nat. Commun.* **4**, 1624 (2013).
10. Quoc An, V. et al. Two-terminal floating-gate memory with van der Waals heterostructures for ultrahigh on/off ratio. *Nat. Commun.* **7**, 12725 (2016).
11. Chen, X. Y. et al. Wafer-scale functional circuits based on two dimensional semiconductors with fabrication optimized by machine learning. *Nat. Commun.* **12**, 5953 (2021).
12. Wang, Y. et al. An in-memory computing architecture based on two-dimensional semiconductors for multiply-accumulate operations. *Nat. Commun.* **12**, 3347 (2021).
13. Migliato Marega, G. et al. Logic-in-memory based on an atomically thin semiconductor. *Nature* **587**, 72-77 (2020).
14. Zhao, F. H. et al. In situ growth of graphdiyne on arbitrary substrates with a controlled-release method. *Chem. Commun.* **54**, 6004-6007 (2018).
15. Wang, H. et al. A sandwich-type photoelectrochemical sensor based on tremella-like graphdiyne as photoelectrochemical platform and graphdiyne oxide nanosheets as signal inhibitor. *Sensor. Actuat. B: Chem.* **304**, 127363 (2020).
16. Zhang, Y. et al. 2D graphdiyne oxide serves as a superior new generation of antibacterial agents. *iScience* **19**, 662-675 (2019).
17. Wu, H. et al. Interfacial charge behavior modulation in perovskite quantum dot-monolayer MoS₂ 0D-2d mixed-dimensional van der Waals heterostructures. *Adv. Funct. Mater.* **28**, 1802015 (2018).
18. Kim, S. et al. In situ observation of resistive switching in an asymmetric graphene oxide bilayer structure. *ACS Nano* **12**, 7335-7342 (2018).
19. Kim, S. K. et al. Conductive graphitic channel in graphene oxide-based memristive devices. *Adv. Funct. Mater.* **26**, 7406-7414 (2016).
20. Hong, S. K., Kim, J. E., Kim, S. O. & Cho, B. J. Analysis on switching mechanism of graphene oxide resistive memory device. *J. Appl. Phys.* **110**, 044506 (2011).
21. Rani, J. R., Oh, S. I., Woo, J. M. & Jang, J. H. Low voltage resistive memory devices based on graphene oxide-iron oxide hybrid. *Carbon* **94**, 362-368 (2015).
22. Fu, T. D. et al. Bioinspired bio-voltage memristors. *Nat. Commun.* **11**, 1861 (2020).

REVIEWERS' COMMENTS

Reviewer #2 (Remarks to the Author):

The authors answered to my comments and suggestions.

Reviewer #3 (Remarks to the Author):

The authors have provided a thorough and detailed reply to my comments and suggestions. The clarity of the work has been significantly improved. The analysis and characterization of GDY and GDYO thin films has strengthened the scientific merit of the manuscript. I do not believe that filament formation is fully understood in GDYO, as there is no experimental data concerning filament size. However, elucidating the microscopic structure of conductive filaments is beyond the scope of the present work.

I believe the manuscript (and supplemental material) is much improved and I strongly recommend publication in Nature Communications, as this work reports a timely and significant advance in high-speed, low-energy, non-volatile memory device development with robust experimental support.

REVIEWERS' COMMENTS

Reviewer #2 (Remarks to the Author):

The authors answered to my comments and suggestions.

Reply: We are grateful for the Reviewer's valuable comments and suggestions for our manuscript. The manuscript does not need to be revised.

Reviewer #3 (Remarks to the Author):

The authors have provided a thorough and detailed reply to my comments and suggestions. The clarity of the work has been significantly improved. The analysis and characterization of GDY and GDYO thin films has strengthened the scientific merit of the manuscript. I do not believe that filament formation is fully understood in GDYO, as there is no experimental data concerning filament size. However, elucidating the microscopic structure of conductive filaments is beyond the scope of the present work.

I believe the manuscript (and supplemental material) is much improved and I strongly recommend publication in Nature Communications, as this work reports a timely and significant advance in high-speed, low-energy, non-volatile memory device development with robust experimental support.

Reply: We are grateful for the Reviewer's valuable comments and suggestions for our manuscript. The manuscript does not need to be revised.